# The Economic Burden and Determinant Factors of Parents/Caregivers of Children with Cerebral Palsy in Malaysia: A Mixed Methods Study

**DOI:** 10.3390/ijerph19010475

**Published:** 2022-01-01

**Authors:** Aniza Ismail, Ruhana Sk Abd Razak, Leny Suzana Suddin, Aidalina Mahmud, Sazlina Kamaralzaman, Ghazali Yusri

**Affiliations:** 1Faculty of Medicine, Universiti Kebangsaan Malaysia, Kuala Lumpur 56000, Malaysia; aniza@ppukm.ukm.edu.my; 2Faculty of Medicine, Universiti Teknologi MARA, Sungai Buloh 47000, Malaysia; leny@uitm.edu.my; 3Faculty of Medicine & Health Sciences, Universiti Putra Malaysia, Serdang 43400, Malaysia; aidalina@upm.edu.my; 4Faculty of Health Sciences, Universiti Kebangsaan Malaysia, Kuala Lumpur 50300, Malaysia; sazlina@ukm.edu.my; 5Academy of Language Studies, Universiti Teknologi MARA, Shah Alam 40450, Malaysia; ghazaliy@uitm.edu.my

**Keywords:** economic burden, Malaysia, cerebral palsy, caregivers, person with disabilities, mixed-method

## Abstract

The economic burden is a major concern for parents/caregivers of children with cerebral palsy (CP). This study used the sequential explanatory mixed-method approach to explorethe economic burden on parents/caregivers with a CP child in Malaysia and the factors associated with the economic burden. The study period spanned April 2020 and December 2020. A total of 106 questionnaire respondents were selected for the quantitative part, and 15 were interviewed to obtain qualitative input. A retrospective costing analysis was conducted based on the cost data obtained from the questionnaire. The majority of the children were GrossMotor Function Classification System (GMFCS) Level 5 (71%), quadriplegic (63%), and aged >4 years (90%). The estimated annual median total economic burden on the parents/caregivers per child in 2020 was RM52,540.00 (~USD12,515.03), with indirect cost being the greatest cost (RM28,800.00, ~USD6860.16), followed by developmental cost (RM16,200.00, ~USD3858.84), direct healthcare cost (RM4540.00, ~USD1081.43) and direct non-healthcare cost (RM3000.00, ~USD714.60). The annual household income was identified as a significant determinant factor (*p*=0.019, 95% CI: 0.04, 0.40) of the economic burden. The participants’ responses during the in-depth interview in the qualitative part of the study supported the premise that socioeconomic factors play a substantial role in determining the total economic burden. Our findings may aid local policymakers when planning the greater provision of support to the affected families in the future, especially for the parents/caregivers of children with CP, who are facing socioeconomic challenges.

## 1. Introduction

Cerebral palsy (CP) is the most common cause of significant motor impairment in children [1]. Despite being the most significant cause of major motor impairment in childhood, CP remains a relatively rare condition, with a birth prevalence of approximately 2 per 1000 births [2]. However, CP and its associated conditions can impose a significant economic burden on the affected families, the health care system, and the general economy.

CP is a common cause of disability in childhood, as it comprises a range of non-progressive syndromes of posture and motor impairment. Over a billion people, i.e., about 15% of the world’s population, have some form of disability [3]. Most CP patients are also considered people with disabilities (PWD) and are largely absent from mainstream health research, experience poorer health, have a greater incidence of chronic conditions and higher health care expenditure than people without disabilities [4]. The management options include physiotherapy, occupational and speech therapy, orthotics, device-assisted modalities, pharmacological intervention and orthopedic and neurosurgical procedures [5]. With their limited movement and comorbid issues, CP patients are required to undergo rehabilitation and treatment services for a long period [5,6,7,8].

PWD, including CP patients, have less access to health care services and, therefore, experience unmet health care needs. In comparison to children with other disabilities, parents of children with learning and multiple disabilities need more support to receive proper childcare [9]. Services for children with disabilities should meet their families’ needs, as the condition affects not only the child but also the caregiver’s health and well-being.

In Malaysia, the CP child can legally be registered as a PWD based on the Persons with Disabilities Act 2008 definition. All PWD can be registered with the Department of Social Welfare Malaysia (also known as Jabatan Kebajikan Masyarakat, JKM) under the Ministry of Women, Family, and Community Development. This registration system requires the PWD themselves to register at designated centers. Upon registration, they will be eligible to utilize the facilities and privileges under the department’s jurisdiction. The facilities and privileges cover a wide range of benefits, including education, financial exemptions and assistance, job support, and outlet discounts. Under the national education system, as students with special needs, children with CP may have access to special education programs based on their ability level [10]. In terms of financial aid, the amount of monetary aid is similar across all categories of PWD and is determined by the government. In the program for PWD, the objectives are focused on assisting them in being self-reliant and achieving their full potential. However, there is no separate category for CP identification to differentiate the type of facility or aid under this department. The distribution of financial aid is also not observed for CP as a separate group, rendering it difficult to ascertain the programs’ magnitude of benefit for people with CP [11]. Regarding the healthcare expenditure incurred by people with CP for healthcare services utilization, all registered PWD are given exemptions from service and health insurance charges limited to outpatient services at government healthcare facilities. Besides this, other types of assistance required by people with CP specifically in society are supported by registered non-governmental organizations such as MyCP (Malaysian Advocates for Cerebral Palsy), GAPS (Gabungan Anak-Anak PalsiSerebrum, Alliance of Children with Cerebral Palsy), and the Kuala Lumpur Cerebral Palsy Association.

Families continue to share the financial burden of childhood disability unevenly. Low-income families are especially vulnerable to burdensome out-of-pocket expenses. Additional efforts are needed to protect these high-risk families [12]. There have been various studies and reports worldwide on the economic burden or lifetime cost of CP, including healthcare, social care, and productivity costs [13,14,15]. However, studies on the financial burden on the parents/caregivers of children with CP according to several determinant factors are still lacking in Malaysia.

Cost of illness (COI) studies are often undertaken to estimate the size and nature of these impacts and are sometimes performed parallel to burden of disease (BOD) studies. BOD studies focus on the incidence/prevalence of the underlying illness and the associated risk factors [16]. The economic burden of CP can be estimated as the lifetime cost for new cases such as incidence costing, as an annual cost for all existing cases of CP as in prevalence costing, or sometimes as a cost for a specific period, condition, or patient subgroup, also termed targeted costing [17].

In 2013, a local COI study on CP showed that direct healthcare cost represents the highest cost, at an annual total of RM14,715.49, which was equivalent to USD4419.06 (2013 average exchange rate: USD0.3003 per RM1). This was followed by developmental costs, direct non-healthcare costs, and indirect costs. The total annual financial cost to care for a child with CP in Malaysia in 2013 was RM29,710.76 (~USD8922.14), which is quite a burdensome amount for the parents/caregivers [6].

A child with CP incurs a mean total average lifetime healthcare cost of USD22,143, compared to a child without any lifetime healthcare needs, who only incurs costs of USD1729, which is approximately 13 times lower [8]. In South Korea, the lifetime healthcare cost of a child with CP was USD26,383, equal to 1.8 times the basic lifetime healthcare cost of the general population, or USD14,579 [7]. In China, a 2003 study reported that the average lifetime economic burden for a CP case was USD67,044 [14]. A study of the economic burden in Denmark indicated that every CP case in the country requires an average lifetime expenditure of €860,000 (~USD1484,344) for men and €800,000 (~USD1595,680) for women [15].

The sociodemographic factors include age, sex, and ethnicity. A local study performed in 2013 found that the economic burden of children with CP was highest in the 7–12-year age group, followed by that in the 13–18-, 4–6-, and 0–3-year age groups [6]. Comparison of the direct healthcare cost of CP patients according to children’s age groups showed the highest-burden in the 7–12-year age group in Malaysia. In comparison, similar studies performed in the United States of America, Denmark, South Korea, and China reported that children in the 0–6-year age group bore the highest direct medical cost [6,7,14,15]. Studies in Japan, Europe, Australia, and Malaysia have reported that the percentages of males with CP exceeded that of females [1,6,15]. Meanwhile, a local study found that Malays comprised the majority of the country’s population of children with CP (87.9%), followed by Chinese (6.8%) and other ethnicities (5.4%) [6].

The severity of illness in children with CP may also affect financial costs, and the lifetime costs may vary significantly with severity, as stated in a study performed in Denmark [15]. A study in Australia concluded that increased CP severity was associated with more hospital admissions and a higher proportion of admissions attributable to respiratory illness, which equals to increased financial cost of illness [18]. Furthermore, another study reported that there was a strong positive relationship between CP severity and expenditure [17]. Regarding socioeconomic factors, low socioeconomic status, defined as a monthly household income of RM1000 or less, may add to parents’ difficulties in finding appropriate and feasible services for children with disabilities, especially older children [19]. A study performed in Malaysia found that the below-average household income group bore a direct medical cost 1.55 times greater than the above-average household income group [6].

Identifying the determining factors influencing the financial burden on parents/caregivers of children with CP may guide the addressing of significant barriers to healthcare this group faces and may aid local policymakers in planning effective service provision to suit the parents’/caregivers’ needs. Thus, in the present study, besides the main objective of estimating the economic burden on parents/caregivers in raising a child with CP in Malaysia, we also studied the determinant factors associated with expenditure, including the child’s and parents’/caregivers’ sociodemographic, the child’s severity of illness, and the parents’/caregivers’ socioeconomic factors. 

## 2. Materials and Methods

### 2.1. Study Design

This was a mixed-method sequential explanatory study involving the sequential collection and analysis of quantitative and qualitative data [20]. Respondents were selected to answer an online questionnaire for the quantitative part; subsequently, respondents who participated in the quantitative part were selected for interviews to gain qualitative input. The study was conducted from April 2020 to December 2020.

### 2.2. Sample Size

The sample size was calculated using the OpenEpi software formula for proportion: [*n* = [DEFF*Np(1 − p)]/[(d^2^/Z^2^_1−α/2_*(N − 1) + p*(1 − p)] using the estimated population number of children with disability in Malaysia data until 2017, i.e., 453,258 [21], the estimated prevalence of moderate CP among children with disability in Malaysia for 2012, i.e., 7.7% [22], design effect of 0.5, precision at 5%. The minimum sample size calculated for a 95% confidence level was 55. After considering an attrition rate of 40% in view of the data being collected using an online survey, the final minimum sample size calculated was 77.

### 2.3. Study Instruments

For the quantitative part, data were collected using a validated, self-administered questionnaire in the Malay language. The questionnaire was developed by a group of experts in public health and early childhood education development [6]. It was created using an online form, specifically Google Forms, and was pre-tested to ensure its reliability. The questionnaire was distributed to the participants via social media platforms, i.e., the MyCP (Malaysian Advocates for Cerebral Palsy) Facebook and WhatsApp groups. The questionnaire comprised 5 sections: Section 1: respondent’s socio-demographic profile, Section 2: child’s demographic profile, Section 3: monthly household expenses data, Section 4: yearly household expenses data, and Section 5: Productivity loss data. 

For the qualitative part, a set of semi-structured guiding questions was created for the in-depth interviews, based on significant contributing factors identified from the quantitative part. These questions were validated by 3experts from the Obstetrical and Gynecological Society of Malaysia (OGSM) and the Perinatal Society of Malaysia. A pilot study was also performed for face validity.

The main driver of the in-depth interviews was to support the quantitative study findings and explore any other issues that was not assessed earlier. To gain more understanding of the factors, the main variables stated in the quantitative part were incorporated in the interview questions, such as the respondents’ sociodemographic details and whether the presence of any other family members with chronic disease or disability affected their expenses. The impact of the respondents’ socioeconomic status on their economic burden and care for their child with CP was also assessed. As a result, the in-depth interview featured 5 semi-structured, open-ended questions:Do you have other family members with chronic disease or disability, and how does this condition affect your expenses?Do you have other family members who can help you care for your special child(ren)?How far can your whole family income extend to afford your special child(ren)’sneeds such as rehabilitation, diet, equipment, health care, and education?Is there any financial issue that prevents you from sending your special child(ren)to school or from obtaining daily necessities?What are the most crucial items our government needs to support in order for you to obtain proper care for your special child(ren)?

### 2.4. Study Population and Sampling

The participants of the quantitative part comprised a convenience sample of parents/caregivers of a child with CP; the majority were members of the MyCP association, a nationwide association for people with CP in Malaysia. Participants were eligible if they agreed to participate in the study, self-reported as the parents/caregivers of a child with CP, with families from or residing in Malaysia, were able to speak and write in Malay, plus were able to access the Internet and the social media platforms used for answering the questionnaire. The exclusion criteria were family with >1 child with CP or family member with disabilities, and families with >1 child or family members with chronic illness and requiring long-term care. Purposive sampling was used for the qualitative part of the study, whereby the sample was selected from respondents in the quantitative part, and sampling was continued until data saturation was reached.

From the targeted sample size of 77, 106 participants enrolled and answered the questionnaire, of whom 15 were selected for further in-depth interviews based on the criteria of age and severity level. Following the data collection, a retrospective costing analysis was conducted using the cost data obtained from the completed questionnaire to estimate the national economic burden on parents/caregivers of children with CP for the entire country, with qualitative input from the interview adding depth to the quantitative analyses results.

### 2.5. Study Variables

The dependent variable in this study was the economic burden on parents/caregivers of children with CP, calculated based on 4cost components: direct healthcare, direct non-healthcare, developmental, and indirect costs.

The direct healthcare cost generally refers to the costs borne by care providers/funders in the health care system and by the individual patient and their family/carers [17]. Direct healthcare costs are expenditures for medical treatment sources used for treating or overcoming the patient’s condition complications, such as outpatient services, warding fees, diagnoses, medical treatment, surgery, medication, and rehabilitation services. In the present study, the direct healthcare cost was divided into rehabilitation services, alternative treatments, medicine costs, medical aid purchasing costs, diagnostic test costs, ward entrance fees, and surgery and consultation fees, as used in a previous local study [6].

Direct costs can sometimes extend to non-health care costs, where education expenses and home modifications are relevant [17]. Direct non-healthcare costs may include transportation and accommodation costs related to the usage of medical services and facilities. Developmental costs refer to expenditures for special needs education, special diet, supplementary diet, and CP childcare services [6,14]. Meanwhile, the indirect cost consists of the productivity loss of a CP child’s parents and of the child themselves. In COI studies, the indirect cost usually refers to productivity losses resulting from morbidity, such as visits to medical practitioners, hospitalization, and premature retirement, together with premature mortality. Productivity losses are based on the impacts assessed for the paid workforce, but can extend to include impacts on domestic production [17].

There were 4independent variables in this study. The first was the CP child’s socio-demographics, which included age, sex, and ethnicity. Age was taken based on the child’s date of birth and categorized by year: 0–3, 4–6, 7–12, and ≥13 years old, which was based on categories used in a local 2013 study [6]. This rendered comparisons between both studies easier. Furthermore, the children’s ages were grouped based on their schooling age. In Malaysia, the majority of children aged 0–3 years are still under the care of their parents/caregivers, and some are sent to nurseries when their parents/caregivers go to work. The standard agerange for children to enter preschool or kindergarten is 4–6 years. Children are required to attend primary school when they are 7–12 years old, while children aged ≥13 years are required to attend secondary school. Sex referred to a state of being male or female in the biological sense and was categorized as male or female. Ethnicity referred to the category of people who share certain inherited physical characteristics, cultures, or beliefs in the social sense and was categorized as Malay, Chinese, Indian, or other.

The second factor was CP severity. The patient’s severity of illness aspect includes the degree of CP severity. Here, CP severity was based on the Gross Motor Function Classification System (GMFCS), a functional classification that differentiates children with CP based on the child’s gross motor function, specifically self-initiated movements, and in particular, sitting and walking, according to five levels of function, starting from Level 1, indicating independent movement, to Level 5, indicating complete assistance required [23]. A study in Denmark that assessed the impact of CP severity on financial costs showed that lifetime cost varies significantly with severity [15]. A study in Australia concluded that increased CP severity and complexity were associated with more admissions and a higher proportion of admissions attributable to respiratory illness [18].

The third factor was the sociodemographic of the parents/caregivers of the children with CP, which also involved age, sex, and ethnicity. Age was categorized into 18–25 years, 36–55 years, and >55 years. The categories started with the nation’s age of majority, i.e., 18 years, the 18–35-year age group for young adulthood, the 36–55-year age group for middle adulthood, and >55 years for those who had reached retirement age and were allowed to fully withdraw their retirement savings [24,25].

The fourth factor studied as an independent variable in this study was the socioeconomics of the parents/caregivers of children with CP, i.e., the household income. We categorized household income based on the average Malaysian household income reported in the 2019 Household Income and Basic Amenities Survey Report by the Department of Statistics Malaysia (DOSM), whereby the average Malaysian household income was RM7901 (~USD1908.09) while the median national income was RM5,873 (~USD1418.33) [26]. An above national average income was defined as an income of >RM7901, while a below national average income was defined as an income of <RM7901. 

To gain a greater understanding of the factors, the main variables above, i.e., the respondents’ sociodemographic details and whether the presence of any other family members with chronic disease or disability affected their expenses, were incorporated in the interview questions. The impact of the respondents’ socioeconomic status on their economic burden and care for their children with CP was also assessed.

### 2.6. Data Collection

For the quantitative part of the study, between April and December 2020, 106 selected parents/caregivers of children with CP completed the online validated Malay version of the self-administered questionnaire distributed via the MyCP Facebook and WhatsApp groups. The participants took approximately <30 min to complete the questionnaire. For the qualitative part of the study, selected respondents were contacted via phone call and invited to participate in an in-depth interview on an agreed-upon date if they agreed to participate.

### 2.7. Data Analysis

For quantitative analysis, the data obtained were initially compiled using a Microsoft Excel spreadsheet before being converted into an SPSS data file. Statistical analyses were performed using IBM Statistical Package for Social Sciences (SPSS) version 23. Univariate analysis was conducted to illustrate the participants’ characteristics and to tabulate the economic burden based on the costing components.

In bivariate analysis, the differences and significances of the economic burden between the categorical independent variables were evaluated with nonparametric tests. The association between economic burden and independent variables consisting of 2categories, or binaries, such as the child’s sex and the annual household income, was calculated using the Mann–Whitney U test. The association between the economic burden and independent variables consisting of ≥3 categories, such as the age group and the CP severity, was calculated using the Kruskal–Wallis test.

The factors associated with total economic burden were examined using multiple linear regression analysis, whereby *p*<0.05 was referred to as the level of significance. The factors studied using this analysis were the child’s age group, sex, comorbidities, and CP severity; and the parents’/caregivers’ age group, sex, and socioeconomic factor, i.e., annual household income.

For qualitative data analysis, a thematic analysis was conducted on the interview data; we discussed the themes and triangulated key themes with quantitative data. All the data were transcribed verbatim. 

For the qualitative data analysis, we used a combination of inductive and deductive approaches to derive the main themes in the thematic analysis. The inductive approach was based on the qualitative data accumulated from the in-depth interview. The deductive approach used predetermined areas of discussion based on known challenges faced by the parents/caregivers of CP children, which we identified from the literature manually and independently. Among the predetermined areas of discussion were issues seeking childcare, the cost of childcare, education for children with CP, government support, and socioeconomic influence on the total economic burden. The inductive approach involved reflexive thematic analysis, while the deductive approach used codebook thematic analysis [27].

## 3. Results

### 3.1. Univariate Analysis

#### 3.1.1. Average Family Economic Burden Categories

Table 1 shows that the median total economic burden on families with CP children in Malaysia in 2020 was RM52,540.00 (~USD12,515.03). The indirect cost, which included parents’ and child’s productivity loss, was the greatest burden, with the median productivity loss being RM28,800.00 (~USD6860.16), which translated to a monthly loss of approximately RM2400.00 (~USD571.68) in 2020.

The second highest cost incurred by families of children with CP in 2020 was developmental costs (RM16,200.00, ~USD3858.84), followed by direct healthcare costs (RM4540.00, ~USD1081.43) and direct non-healthcare costs (RM3000.00, ~USD714.60). The cost of purchasing dietary supplements (RM6000.00, ~USD1429.20) was the most expensive under the developmental component cost. The domestic helper component was the second-highest cost (RM4800.00, ~USD1143.36), while the daily necessities cost was the third-highest, totaling RM2700.00 (~USD643.14).

A comparison of the components in the direct healthcare cost category showed that the average cost of purchasing assistive devices was the highest (RM2500.00, ~USD595.50). Rehabilitation services (RM1200.00, ~USD285.84) were the second-highest in the direct healthcare cost category, followed by the alternative treatment component (RM840.00, ~USD200.09).

#### 3.1.2. Characteristics of the Children with CP and Their Parents/Caregivers

A total of 106 parents/caregivers of children with CP who had completed the online questionnaire were selected for inclusion in this study. Table 2 shows the results of descriptive analysis based on the data obtained. The average age of the children with CP was 9 years; 67.0% of the 106 children with CP were male. The mean age of the parents/caregivers who responded to the questionnaire was 39.47 years; 76.4% were female. Based on ethnicity, the majority were Malay (87.7%); 0.9% of parents/caregivers were Indian and Chinese, while 10.4% of parents/caregivers were of other ethnicities. Most of the respondents were from the state of Selangor (30.48%), followed by those from Sabah (21.9%), Johor (9.5%), Kedah (7.6%), Perak and Negeri Sembilan (each, 4.8%), the Federal Territory (FT) of Kuala Lumpur and Malacca (each, 3.8%), Penang and Pahang (each, 2.9%), Sarawak and Perlis (each, 1.9%). Respondents were fewest from FT Putrajaya, Terengganu, Kelantan, and FT Labuan (each, 1.0%).

The descriptive statistical analysis for the severity of illness factor showed that most of the children had higher CP severity: 70.8% were categorized with GMFCS Level 5 CP, followed by GMFCS Level 4 (17.9%) and GMFCS Level 3 CP (11.3%). In addition, 67.3% of the children had comorbid illnesses; 32.7% were without comorbid illnesses, showing a minimal difference with a previous study in which 63.5% of CP children had comorbid illness and 36.5% of CP children did not have comorbid illness.

Here, 79.2% of the children received healthcare services mainly from government health clinics and hospitals, while around 58.5% received monetary assistance. Among the sources of monetary assistance were from the JKM, through the Financial Assistance for Carers of Bed-Ridden Individuals with Disability and Chronically Ill Patients (BPT), and the Ministry of Education Malaysia (MOE) Allowance for School Children with Special Needs, which is specifically for school-aged children with disability.

In 2020, the average family household income was RM3286.11/month (~USD782.75/month). Compared to the 2019 average Malaysian household income as reported by DOSM, 92% of the families of children with CP had household incomes below the national’ average, which is below RM7,901.00 (~USD1882.02) [26].

### 3.2. Bivariate Analysis

#### 3.2.1. Comparison of Economic Burden According to Age Group

Table 3 shows that the economic burden on children with CP in the ≥13-year age group was highest at RM78,000.00 (~USD18,579.60), followed by the 4–6-year (RM52,360.00, ~USD12,472.15), 7–12-year (RM45,400.00, ~USD10,814.28), and 0–3-year age groups (RM30,720.00, ~USD7317.50). Meanwhile, among the parents/caregivers, the economic burden on the 36–55-year age group was highest at RM57,520.00 (~USD13,701.26), followed by the 18–35-year (RM49,500.00, ~USD11,790.90) and >55-year age groups (RM39,600.00, ~USD9432.72).

Comparison of direct healthcare cost according to economic burden category showed that it was highest in the >13-year age group (RM5400.00, ~USD1286.28), followed by the 4–6-year (RM5260.00, ~USD1252.93), the 7–12-year (RM4600.00, ~USD1095.72) and the 0–3-year age groups (RM3600.00, ~USD857.52). Direct healthcare cost comparison among the parents/caregivers showed that the 36–55-year age group incurred the highest direct healthcare cost per year (RM5420.00, ~USD1291.04) compared to the younger 18–35-year (RM4800.00, ~USD1143.36) and older age groups (RM1800.00, ~USD428.76).

The economic burden comparison for direct non-healthcare cost showed that the 0–3-year age group recorded the highest amount (RM3600.00, ~USD857.52), followed by the >13-year (RM3000.00, ~USD714.60), 4–6-year (RM2700.00, ~USD643.14) and 7–12-year age groups (RM2400.00, ~USD571.68). The direct non-healthcare cost comparison of the parents/caregivers showed that the 18–35-year age group incurred the highest direct healthcare cost (RM3600.00, ~USD857.52) compared to the 36–55-year (RM2400.00, ~USD571.68) and the >55-year age groups (RM1800.00, ~USD428.76).

For the developmental cost, the ≥13-year group bore the highest cost (RM15,600.00, ~USD3715.92), followed by the 7–12-year (RM14,400.00, ~USD3430.08), 4–6-year (RM12,000.00, ~USD2858.40), and 0–3-year groups (RM8520.00, ~USD2029.46). Among the parents/caregivers, the 36–55-year age group bore the highest cost (RM15,600.00, ~USD3715.92), followed by the 18–35-year (RM13,200.00, ~USD3144.24) and >55-year age groups (RM9300.00, ~USD2215.26).

Among the CP children, the indirect cost was highest for the ≥13-year age group (RM54,000.00, ~USD12,862.80), followed by the 4–6-year (RM32,400.00, ~USD7717.68), 7–12-year (RM24,000.00, ~USD5716.80) and 0–3-year age groups (RM15,000.00, ~USD3573.00). The comparison of the parents/caregivers showed that the 36–55-year age group had the highest indirect cost burden, followed by the 18–35-year and >55-year age groups.

#### 3.2.2. Comparison of Economic Burden According to the Children’s Sex

Table 4 shows the economic burden based on the CP children’s sex, household income, and GMFCS level-based CP severity. The total expenditure of male CP children was RM51,600.00 (~USD12,291.12), which is higher compared than that of female CP children (RM45,500.00, ~USD10,838.10). Comparison of the economic burden categories showed that male CP children had 1.25 times higher expenditures for direct non-healthcare cost, 1.23 times higher developmental cost, and 1.07 times higher indirect cost than female CP children.

Comparison of the direct healthcare cost between the sexes showed that male CP children had double the expenditure for medical aids purchases as compared to female CP children. Meanwhile, female CP children had 1.75 times higher expenditure for rehabilitation services and doubled the alternative treatment cost compared to male CP children. For the developmental cost, male CP children had 1.5 times higher expenditure for daily necessities compared to female CP children, 1.42 times higher expenditure for domestic helpers, and 1.25 higher cost for dietary supplements. Female CP children had 1.74 times higher expenditure for insurance cost than male CP children.

Overall, the comparison shows that male CP children incurred higher expenditure costs compared to female CP children. However, the Mann–Whitney U test showed that there were no significant differences in the economic burden between CP children according to sex.

#### 3.2.3. Comparison of Economic Burden According to Household Income

Table 4 shows that the economic burden on families with above-average household incomes was higher (RM120,164.00, ~USD28,623.06) compared to the below-average household income group (RM46,350.00, ~USD11,040.57). However, this comparison was not significant, with *p*>0.05 using the Mann–Whitney U test.

The highest cost for both household income groups was indirect cost, specifically parents’ and child’s productivity loss, as compared to the direct healthcare and developmental costs. The productivity loss for the above-average household income group (RM93,000.00, ~USD22,152.60) was statistically significantly higher than that of the below-average household income group (RM24,000.00, ~USD5716.80). Medical aids purchases were statistically significantly higher in the above-average household income group (RM6800.00, ~USD1619.76) compared to the below-average household income group (RM2365.00, ~USD563.34).

In brief, the above-average household income group incurred higher direct healthcare, developmental and indirect costs compared to the below-average household income group. Only the direct non-healthcare cost, specifically transportation cost, was higher in the below-average household income group as compared to the above three cost categories. Transportation cost was RM3,300.00 (~USD786.06) for the below-average household income group and RM2400.00 (~USD571.68) for the above-average household income group. However, the comparison was not significant based on the Mann–Whitney U test result, with *p* = 0.219.

#### 3.2.4. Comparison of Economic Burden According to CP Severity

According to the GMFCS level-based severity levels (Table 4), the economic burden on the parents/caregivers of children with CP was higher in families with children with GMFCS Level 3 (RM57,050.00, ~USD13,589.31), compared to those with GMFCS Level 4 and 5 CP, which had total incurred costs of RM50,860.00 (~USD12,114.85) and RM45,400.00 (~USD10,814.28), respectively. The Kruskal–Wallis test showed that there were no significant differences in the economic burden between the GMFCS level-based level of severity.

The indirect cost of raising children with CP was the highest among the cost components studied, followed by the developmental, direct healthcare, and direct non-healthcare costs. The parents’ and child’s productivity loss, calculated under the indirect cost component, was highest in families with children with GMFCS Level 3 (RM31,800, ~USD7574.76), followed by those with Level 4 (RM30,000.00, ~USD7146.00) and Level 5 (RM24,000.00, ~USD5716.80).

Under developmental cost, the highest burden was on the families of children with GMFCS Level 5 CP, followed by those with Level 3 and Level 4. The highest contributing costs were the dietary supplement and domestic helper service costs.

Total costs incurred for direct healthcare cost was highest in families with children with GMFCS Level 3 (RM7700.00, ~USD1,834.14), followed by those with Level 5 (RM4550.00, ~USD1083.81) and Level 4 (RM1875.00, ~USD446.63). For the direct non-healthcare cost, families with children with GMFCS Level 4 bore the highest total transportation cost per year (RM3600.00, ~USD857.52) compared to Level 3 (RM3000.00, ~USD714.60) and Level 5 groups (RM2400.00, ~USD571.68).

### 3.3. Multivariate Analysis

#### Regression Analysis of the Determinant Factors of Economic Burden on Parents/Caregivers

A multiple linear regression was calculated to predict the economic burden on parents/caregivers of children with CP according to several factors: sociodemographic factors (e.g., sex and age group), socioeconomic factors (i.e., annual household income), and the CP severity level. A significant regression equation was found (*F*(4,101) = 2.948, *p* = 0.024), with R^2^ = 0.105. Based on the factors’ unique individual contributions, household income (*β* = 0.234, *t* = 2.392, *p* = 0.019) was a positive predictor of economic burden (Table 5).

The participants’ predicted economic burden was equal to RM24,163.96 + (0.234 × annual household income), whereby the annual household income is measured in Ringgit Malaysia (RM). There was a significant linear relationship between annual household income and parents’/caregivers’ total economic burden. Those with RM1000 higher annual household income had a higher total economic burden of RM234.00 (95% CI: 0.04, 0.4).

Overall, the annual household income factor was statistically significant in determining the economic burden of caring for children with CP as compared to the factors of parent/caregiver age group, child’s sex, and GMFCS level-based CP severity.

### 3.4. Qualitative Findings

The qualitative technique selected for this study was an in-depth interview. Five general interview questions were developed according to the research questions and were validated by experts from the OGSM and the Perinatal Society of Malaysia. 

The interview was conducted among parents/caregivers of CP. A total of 15 respondents were selected for the interview. The respondents participated voluntarily, and all of them have at least one special needs child under GMFCS Level ≥3. Every respondent was labeled accordingly, e.g., R1 refers to respondent 1. The data were transcribed verbatim and analyzed thematically according to the research questions. The main objective was to explore and understand the cost of spending and issues related to the cost of a child with CP, which cannot be clarified via a questionnaire. The following are several themes derived and emerged from the interview session.

Seeking Childcare

The interviews conducted with 15 respondents revealed that most parents/caregivers preferred to care for their children on their own due to several factors. Some of the families with children with CP lived far away from their family members, hence the priority to care for the children themselves. Some were able to request assistance from neighbors if required; however, others received no assistance, including from their own family members. 

“The acceptance from my husband’s side is less compared to my side, but my family stay far from my place. I have a sister stay nearby but she is very busy with her autistic son. That’s why I rely more on myself.”(R1)

“I can leave my daughter to my neighbor. She used to look after her if I need to go out for a while.”(R4)

“In the beginning, there was nobody who can take care of my son.”(R5)

In caring for their own children, especially those with special needs, parents/caregivers may also reduce the extra financial burden of paying for domestic helpers and do not have to worry about the problems that may arise if the child were unwilling to be cared for by strangers, or the low level of confidence in hiring suitable helpers due to the severity of the child’s condition.

“It is difficult to leave my son with others. I have no confidence. His condition is very severe. I am taking care of him 24/7. My husband and I will take turns to take care of him if one of us needs to go out. If somebody else needs to take care of him, it must be a well-trained person. I have searched the cost for it and it is almost RM5000 a month, which I can’t afford as I need to pay for his necessities as well.”(R5)

b.Cost of Childcare

Combined with the need for caring for their special needs children, the high cost of childcare also affected the parents’/caregivers’ socioeconomically. Some had to depend on financial aid such as from the zakat system, a system in the Muslim economy based on the compulsory obligation for Muslims to tithe their wealth at the rate of 2.5% and which is paid to an authorized religious center, to be distributed to eight groups of eligible people termed asnaf [28].

“I received some financial aids from zakat, but it’s still not enough.”(R8)

Some of the respondents’ spouses even had to seek extra income by taking on part-time jobs such as being Grab drivers.

“I sold my spa business, and I did online business and dropship.” (R1)

“My husband needs to work as Grab driver to get additional income. He works until midnight and also on weekends.”(R5)

A few admitted that they were unable to afford several child healthcare service options due to the high cost, such as private rehabilitation services.

“Private rehab is very expensive.”(R6)

Due to a few factors, some parents/caregivers had to send their child to expensive childcare services, despite the extremely high fee, and some even had to limit their own daily food intake to afford the cost of treatment for their child.

“The cost of special childcare center is very high, and it keeps increasing. It’s very expensive, up to RM2000 monthly.”(R6)

c.Education for Children with CP

Aside from the expensive cost of private schools or schools specialized for PWD, some parents/caregivers acknowledged during their interview that daily necessities should sometimes take priority over their child’s higher education cost. Special schools having limited staff to care for their child was another factor in the parents’/caregivers’ selection of their child’s medium of learning.

“I sent my son to a special school and special daycare at the same time. But they only put my son in a baby cot which made my son traumatic and felt terrible. Maybe they have limited staff.”(R6)

“I spent a lot for special needs school. Last time it was RM700 and then the fee increased to RM950.”(R6)

d.Government Support

Based on the input shared by the parents/caregivers during the interviews, besides the need to strengthen the special education system, among the other needs the government should prioritize for children with CP was the need to revise treatment costs, as private healthcare services are very expensive, and around 79.2% of the respondents made government healthcare settings their main healthcare provider of choice. Some of the high treatment costs, such as essential medical aids and rehabilitation services, should also be reviewed so that the government may plan accordingly to reduce the burden.

“I have to reuse some equipment like syringe and bottles to save my expenses.”(R5)

“I don’t have proper wheelchair for my 4 years old special son. I don’t receive any help.”(R8)

Besides transportation aid, the respondents also recommended prioritizing government assistance for utility costs, such as discount rates on monthly bills from utility providers, which may help alleviate their economic burden. Existing financial aid should also be reviewed regularly to account for the rising financial burden on families of children with CP, as well as the impact of changes in the inflation rate and economic trends.

“The government needs to review the eligibility of the family who receive financial aids. The current practice is now irrelevant.”(R1)

“Staff at Welfare Division do not know the cost that we have to pay for therapy…”(R1)

“The government needs to support us to adapt our life. It is very hard.”(R7)

e.Socioeconomic Influence on Total Economic Burden

Financial burdens also vary according to the household socioeconomic status. The multivariate analysis showed that the annual household income factor had a statistically significant impact on the economic burden on parents/caregivers of children with CP. Families with higher annual household income face a greater financial burden. The upper-middle-income households, for instance, may have the financial means to care for special needs children but also falls into the categories receiving less government assistance.

The unstable economy, particularly during the corona virus disease 2019 (COVID-19) epidemic, exposed the parents/caregivers to the possibility of greater productivity loss due to unemployment and loss of income. Moreover, families with financial commitments were more burdened compared to those who had just started their families. Children with CP who require additional attention and care may compel their parents/caregivers to stop working, significantly reducing their household income. On the other hand, if parents/caregivers continue to work, the cost of employing a domestic helper would also increase their financial burden. This may eventually result in mental stress for the parents/caregivers. In general, raising children with CP has long-term effects on not only their socioeconomic situation but also on the family’s mental health.

“There are three types of family. Firstly, new family who has just started their family life. Secondly, old family who depends on two incomes from both husband and wife. Suddenly, they have a special needs child. Thirdly, a family who has only one side works either husband or wife. The impact of economic burden is more severe for the second type of family because they used to have two incomes and used to have certain level of lifestyles and commitments. This actually happened to me.”(R7)

## 4. Discussion

Based on the analysis, the calculated total economic burden on the parents/caregivers of children with CP was RM52,540.00 per year, which is equivalent to USD12,515 per year, using the 2020 average exchange rate of USD0.2382 per RM1.00 as reference. This is higher than the findings of a previous local study, which reported a total cost of financing per year of RM29,710.76 [6]. With the economic strain from various factors, including the coronavirus disease 2019 (COVID-19) pandemic outbreak during the study period and the effects of lockdowns, the amount is a considerable burden on such parents/caregivers, who already face the challenge of providing for a disabled child [29].

The indirect cost, which includes parent and child productivity loss, was the greatest burden. Given that the country was facing lockdowns in response to the COVID-19 pandemic at the time of the study, most businesses implemented cost-cutting initiatives to address the economic effects. Some businesses were even forced to cease operations due to the huge losses incurred [30]. This resulted in a higher number of job losses, which also affected these parents/caregivers.

The next highest financial burden was developmental costs, followed by direct healthcare costs and direct non-healthcare costs. Under the development component cost, the cost of purchasing dietary supplements was the highest and was largely used for buying formula milk, supplementary diets, nasogastric tubes, and milk pumps. The second highest cost was for domestic helper services, which could be high due to the limited availability of home helpers during the COVID-19 lockdowns. Daily necessities cost was the third-highest, where it was needed for purchasing daily necessities, personal care products specialized for sensitive skin, diapers, wet tissues, and other essentials. The average annual cost of special education for children with CP was RM1560.00 (~USD371.59). However, most of the parents/caregivers opted to depend on the Community-Based Rehabilitation (CBR) Programme, more well-known as Program Pemulihan Dalam Komuniti (PDK), for their child’s education, due to the expensive cost of non-government special schools. The PDK is a program created by the JKM Disabled Development Department (JPPWD) and implemented throughout the country through the integrated efforts of PWD and their families; communities and health services; and education, vocational, and social services [31].

The cost of purchasing assistive devices was the highest component under the direct healthcare cost category. Individuals registered as PWDs are entitled to receive financial support from the government, provided via the Procurement of Artificial Limbs and Assistive Devices (BAT/S) scheme [32]. The financial aid eases the burden of purchasing walking aids or other assistive devices for the child. Unfortunately, maintenance services or spare parts must still be paid out-of-pocket, as the scheme does not cover them.

Rehabilitation services ranked second in terms of direct healthcare costs. Government- and community-based rehabilitation centers established across the country provide services predominantly to the low- to medium-socioeconomic status population [21]. Nevertheless, some parents/caregivers may face difficulty adhering to the specified follow-up dates, requiring them to seek alternative rehabilitation services. Complementary and alternative treatment methods (CAM) ranked third under direct healthcare costs, showing the high interest among parents/caregivers to learn more about new treatment options for CP disorders, and are believed to be able to improve the child’s quality of life. The CAM used for CP includes massage therapy, aquatherapy, hippotherapy, acupuncture, and stimulatherapy [33].

Direct non-healthcare cost such as the transportation costs incurred for bringing the child to follow-ups also increase the financial burden, mainly for the lower-income group. The currently available public transport aid, such as concession card discounts, is beneficial for PWD; however, it is not applicable for all targeted persons throughout the country. For example, poor people living in rural highland areas in East Malaysia may be unable to access public transportation services established for city dwellers in Kuala Lumpur, or access public trains, as there are no railways there due to the region’s geography. Additional tax exemptions for PWD or their caregivers specifically for transportation purposes would undoubtedly be a tremendous relief, given there were none when the present study was conducted [34].

The sociodemographic factor of children with CP in the present study is quite similar to that of prior studies, whereby the majority of CP children are male [6,35], reflecting the global prevalence of the male gender in CP. Our results also show that the cost of caring for male children with CP was higher compared to that for female children, which is similar to the findings of a previous study [15].

In the present study, the mean parent/caregiver age was 39.47 years. Being Generation Y (Gen Y), their age may also indirectly reflect their higher financial commitments, as a local economic study revealed that 75.0% of Gen Ys have at least one source of long-term debt, while 37% have more than one long-term debt obligation [36]. Coupled with the extra financial burden of caring for a child with CP, this may lead to economic, mental, and health issues among the affected family members, in addition to experiencing the additional negative effects of the COVID-19 pandemic [29].

Here, the average monthly family household income of families of children with CP was RM3286.11 (~USD782.75), with a monthly median of RM2858.00 (~USD680.78). Compared to the average Malaysian household income, 92% of the families of children with CP in this study had household incomes below the average national household income, more than the 85.1% reported in a 2013 local study [6]. Our results also reveal that the below-average household income group pays RM540.00 (~USD128.63) annually for daily nursery expenses, while none in the above-average household income group acquired the service. However, the above-average household income group had a greater financial capability for hiring home helpers to care for their CP child, which incurred more than five times the cost of daily nursery fees.

The economic burden on the parents/caregivers varied according to CP severity level, with the highest burden on the families of children with GMFCS Level 3 CP, followed by those with GMFCS Level 5 and 4 CP. The majority of parents received healthcare services and education from public facilities. As supported by the interview findings, the majority of parents were unable to afford special child healthcare services and education options such as private rehabilitation services and private schools due to the high cost. The lifetime cost varies significantly with CP severity [15]. Another study concluded that increased CP severity was associated with more hospital admissions [18], leading to a higher economic burden.

Based on the analysis, annual household income was identified as a significant determinant factor of the economic burden (*p* = 0.019, 95% CI: 0.04, 0.40). The qualitative data supported this quantitative data finding. As supported by the participants’ responses during the in-depth interview, socioeconomic factors play a huge role in determining the total economic burden. Thus, local policymakers should prioritize those most impacted when planning more assistance for affected families, particularly the parents/caregivers of children with CP facing socioeconomic challenges.

The five themes that emerged from the in-depth interviews managed to enable a better understanding of the caregivers’ perception of the economic burden in caring for CP children. The qualitative findings show that three of the five themes were related to the financial aspects of caring for a child with CP. The caregivers coped with the financial burden by providing their own care to their child instead of paying for professional care, which in turn could lead to them being unable to go to work. This could explain the quantitative finding that the greatest burden reported by the caregivers was productivity loss. Moreover, even though the government designates financial aid for caregivers under the JKM, which amounts to about RM300 (~USD71.46) per month [32], this value is much lower than the expected fee for hiring a professional carer, which could amount to RM5000 (~USD1191), as one of the respondents stated. When related to the cost from quantitative data analysis, which found that the minimum annual cost for a domestic helper was RM600 (~USD142.92) and that it might go up to RM20,130 (USD4794.97), it appears that the adequacy of assistance varies between caregivers. This contrasting finding raises the possibility that caregivers value the cost of domestic helpers differently. This was supported by the other emerging themes of the cost of childcare, whereby the caregivers needed to earn extra income and not depend on aid from the government or non-governmental organizations. Notably, in this same aspect, the respondents mentioned private service utilization. The high cost was expected because, in Malaysia, PWD are exempted from health service fees only at public facilities and not when they seek treatment at private facilities. The reason for utilizing private facilities was outside the scope of this study and should be considered for further exploration in the future. Additionally, in terms of financial aspects, the caregivers, the themes on socioeconomic influence on the total economic burden, and the caregivers’ perception raised interesting points on whether this situation can be interpreted as inequity in the care of CP patients in the big picture. However, as there are no standard guidelines on optimal care for CP patients in Malaysia, the difference in caregiving style and caregivers’ coping mechanisms requires exploration before conclusions can be drawn to explain the issue of inequity.

The remaining two themes of the qualitative part could be interpreted as caregivers’ expectations of the role policymakers, and the government should play in assisting the vulnerable CP population. The theme on education for children with CP reflected the caregivers’ fear and sacrifices in ensuring that their CP child did not miss out on obtaining an education. The findings raise not only the question of availability but also the safety aspect when a CP child enters school. In Malaysia, the government had shown commitment in this regard, as explicitly translated in the Malaysia Education Blueprint 2013–2025, with aims to achieve 75% enrolment of students with special needs in inclusive programs by 2025 to ensure access to education for all children, including those with disabilities [37]. Thus, it is expected that 25% of children will still encounter problems accessing the related programs, which supports this qualitative theme finding. The other related theme in this aspect was on government support, whereby caregivers shared their expectations of the government to provide a mechanism to manage the market price of items utilized for the care of CP patients and as the authority figure to be up-to-date with the increasing expenses for caring for CP patients and to be able to provide appropriate and timely assistance for the family or patients with CP. This theme supports the quantitative finding on the increased annual total cost in the present study, which is almost double that of a 2018 study [6]. To reiterate, although financial assistance is currently available under the JKM, the amount of aid will depend on the government’s budgetary policy. In the National Budget Year 2021 allocation, the Malaysian government has shown its commitment to assisting the vulnerable population in terms of monetary assistance, where financial aid in the form of allowance for the PWD category was increased by 13–43% its original amount [38].

Our study has several limitations. The sample size is small, which may have resulted in larger standard errors and CIs. The samples were obtained from parents/caregivers who were members of an association and had access to mobile phones. Hence, they may not have included non-association member parents/caregivers or those who do not have mobile phone access.

## 5. Conclusions

Overall, there is an increasingtrend in the estimated total economic burden on parents/caregivers for raising a child with CP in Malaysia, totaling around RM52,540.00 (~USD12,515) in 2020, with indirect cost that includes parent and child productivity loss contributing the largest share of the financial burden. Several factors influencing the financial burden were explored, and the socioeconomic status of the affected families was identified as a significant determinant factor. The results indicate that financial aid should not be limited to low-income groups, as a large proportion of middle-income families also struggle financially to care for children with CP.

Numerous inputs from the parents/caregivers involved in this study also address the recommendations on the support the government should prioritize to alleviate the economic strain on this affected population. We hope that this study can assist local policymakers as well as the relevant government agencies and non-governmental organizations in providing greater support to the affected families in the future. The development and implementation of integrative public policies may expedite the provision of assistance.

Policy development could start with national-level inter-agency dialogue for devising vital strategies that will not only be able to relieve the burden of caregivers and CP patients but also to search for means of reducing the incidence of CP in the future. Three bases for strategies that can be highlighted from this study are: improving access and adequacy of financial aid, anticipating the progressive health needs related to the condition, and lifetime education opportunities to ensure that CP patients are not overlooked in terms of the opportunity to participate in society.

Nevertheless, there should be more research on the needs, barriers, and health outcomes for children with CP. Strengthening the healthcare system may improve the care and health outcomes for children with disabilities. Further local studies are highly recommended, particularly those that encompass a larger and more diversified portion of the Malaysian population, as the number of local studies on this subject is quite limited.

## Figures and Tables

**Table 1 ijerph-19-00475-t001:** Parents’/caregivers’ economic burden per year.

Economic Burden Category/Financial Cost Components		Min.		Max.		Mean		SD		Median		IQR	
		RM	USD	RM	USD	RM	USD	RM	USD	RM	USD	RM	USD
Direct Healthcare Cost	Rehabilitation Services	0.00	0.00	18,000.00	4287.60	2563.26	610.57	3495.66	832.67	1200.00	285.84	857.52	204.26
	Alternative Treatments	0.00	0.00	18,000.00	4287.60	1519.52	361.95	2471.18	588.64	840.00	200.09	571.68	136.17
	Medicine Costs	0.00	0.00	6000.00	1429.20	871.67	207.63	1262.59	300.75	0.00	0.00	285.84	68.09
	Purchase of Medical Aids	0.00	0.00	31,300.00	7455.66	3925.80	935.13	4789.71	1140.91	2500.00	595.50	976.62	232.63
	Diagnostic Tests	0.00	0.00	15,000.00	3573.00	379.41	90.38	1888.84	449.92	0.00	0.00	0.00	0.00
	Ward Entrance Fees/Surgery	0.00	0.00	75,000.00	17,865.00	2050.81	488.50	10,573.97	2518.72	0.00	0.00	35.73	8.51
	Consultation Fees	0.00	0.00	4200.00	1000.44	97.59	23.25	584.84	139.31	0.00	0.00	0.00	0.00
	Total	0.00	0.00	167,500.00	39,898.50	11,408.05	2717.40	25,066.79	5970.91	4540.00	1081.43	2727.39	649.66
Direct Non-Healthcare Cost	Transportation Costs	0.00	0.00	30,000.00	7146.00	4289.40	1021.74	4735.24	1127.93	3000.00	714.60	857.52	204.26
	Total	0.00	0.00	30,000.00	7146.00	4289.40	1021.74	4735.24	1127.93	3000.00	714.60	857.52	204.26
Developmental Cost	Domestic Helper	600.00	142.92	20,160.00	4802.11	6494.40	1546.97	4767.05	1135.51	4800.00	1143.36	1214.82	289.37
	Nursery	0.00	0.00	18,000.00	4287.60	2737.89	652.17	3969.16	945.45	0.00	0.00	1143.36	272.35
	Special Education	0.00	0.00	12,000.00	2858.40	1560.00	371.59	2792.58	665.19	0.00	0.00	571.68	136.17
	Dietary Supplements	0.00	0.00	24,000.00	5716.80	6670.60	1588.94	4689.67	1117.08	6000.00	1429.20	1143.36	272.35
	Daily Necessities	600.00	142.92	24,000.00	5716.80	4240.47	1010.08	4010.13	955.21	2700.00	643.14	1014.73	241.71
	Insurance	50.00	11.91	8500.00	2024.70	1942.41	462.68	2122.19	505.51	1500.00	357.30	524.04	124.83
	Others	0.00	0.00	18,000.00	4287.60	2114.75	503.73	3272.81	779.58	1200.00	285.84	643.14	153.20
	Total	1250.00	297.75	124,660.00	29,694.01	25,760.52	6136.16	25,623.58	6103.54	16,200.00	3858.84	6255.13	1489.97
Indirect Cost	Parents and Child Productivity Loss	3000.00	714.60	102,000.00	24,296.40	32,477.14	7736.05	22,888.34	5452.00	28,800.00	6860.16	7003.08	1668.13
	Total	3000.00	714.60	102,000.00	24,296.40	32,477.14	7736.05	22,888.34	5452.00	28,800.00	6860.16	7003.08	1668.13
Overall Total Economic Burden/Year				424,160.00	101,034.91	73,935.11	17,611.35			52,540.00	12,515.03		

Min. = Minimum. Max. = Maximum. IQR = Interquartile range. RM = Ringgit Malaysia. USD = United States Dollar.

**Table 2 ijerph-19-00475-t002:** Characteristics of children with CP and their parents/caregivers.

Characteristic	Median (IQR)	Mean (SD)
Child’s age (year)	7 (7)	9 (5.65)
Parents’ age (year)	37 (12)	39.47 (7.99)
Household income/month (RM)	2858.00 (2500)	3286.11 (2650.31)
Household income/month (USD)	680.78 (595.50)	782.75 (631.30)
Characteristic	N	Percentage (%)
Child’s Age Distribution (Years)		
0–3	11	10.4
4–6	35	33.0
7–12	35	33.0
≥13	25	23.6
Child’s Sex		
Male	71	67.0
Female	35	33.0
Child’s Ethnicity		
Malay	93	87.7
Indian	1	0.9
Chinese	1	0.9
Other	11	10.4
State of Residence		
Perlis	2	1.9
Kedah	8	7.6
Pulau Pinang	3	2.9
Perak	5	4.8
Selangor	32	30.5
W.P. Putrajaya	1	1.0
Negeri Sembilan	5	4.8
Melaka	4	3.8
Johor	10	9.5
W.P. Kuala Lumpur	4	3.8
Pahang	3	2.9
Terengganu	1	1.0
Kelantan	1	1.0
Sabah	23	21.9
Sarawak	2	1.9
W.P. Labuan	1	1.0
Child’s GMFCS Level		
3	12	11.3
4	19	17.9
5	75	70.8
Child’s CP Classification		
Quadriplegia	67	63.2
Diplegia	13	12.3
Hemiplegia	3	2.8
Unsure/Other	23	21.7
Comorbid Grouping of Children with CP		
Without comorbid illness	33	32.7
With comorbid illness	68	67.3
Parents’/Caregivers’ Age Distribution (Years)		
18–35	41	40.2
36–55	59	57.8
>55	2	2.0
Parents’/Caregivers’ Sex		
Male	25	23.6
Female	81	76.4
Family Annual Household Income		
Below average of Malaysian household income	92	92.0
Above average of Malaysian household income	8	8.0
Parents/Caregivers’ Dependents (n)		
<5	80	75.5
≥5	26	24.5
Main Healthcare Provider of Choice		
Government hospitals/clinics	84	79.2
Non-government healthcare providers	22	20.8
Monetary Assistance		
Not receiving any monetary assistance	44	41.5
Received any monetary assistance	62	58.5

IQR = Interquartile range.SD = Standard deviation.N = Number of subjects.RM = Ringgit Malaysia. USD = United States Dollar.

**Table 3 ijerph-19-00475-t003:** Comparison of CP economic burden according to age group.

Economic Burden Category	Age Group of Children with CP (Years)									Age Group of Parents/Caregivers (Years)						
	0–3		4–6		7–12		≥13		Kruskal– Wallis ^a^	18–35		36–55		>55		Kruskal– Wallis ^a^
	RM	USD	RM	USD	RM	USD	RM	USD		RM	USD	RM	USD	RM	USD	
Direct Healthcare Cost																
Rehabilitation Services	2880.00	686.02	1200.00	285.84	1680.00	400.18	600.00	142.92	0.260	1200.00	285.84	2160.00	514.51	0.00	0.00	0.289
Alternative Treatments	0.00	0.00	1200.00	285.84	600.00	142.92	1020.00	242.96	0.686	1200.00	285.84	1200.00	285.84	0.00	0.00	0.286
Medicine Costs	0.00	0.00	0.00	0.00	0.00	0.00	1200.00	285.84	0.039 *	0.00	0.00	0.00	0.00	1200.00	285.84	0.406
Purchase of Medical Aids	465.00	110.76	2365.00	563.34	3400.00	809.88	3700.00	881.34	0.085	1865.00	444.24	3000.00	714.60	600.00	142.92	0.140
Diagnostic Tests	0.00	0.00	0.00	0.00	0.00	0.00	0.00	0.00	0.568	0.00	0.00	0.00	0.00	0.00	0.00	0.117
Ward Entrance Fees/Surgery	750.00	178.65	0.00	0.00	0.00	0.00	0.00	0.00	0.037 *	0.00	0.00	0.00	0.00	0.00	0.00	0.246
Consultation Fees	0.00	0.00	0.00	0.00	0.00	0.00	0.00	0.00	0.828	0.00	0.00	0.00	0.00	-	-	0.352
Total	3600.00	857.52	5260.00	1252.93	4600.00	1095.72	5400.00	1286.28	0.810	4800.00	1143.36	5420.00	1291.04	1800.00	428.76	0.167
Direct Non-Healthcare Cost																
Transportation Costs	3600.00	857.52	2700.00	643.14	2400.00	571.68	3000.00	714.60	0.867	3600.00	857.52	2400.00	571.68	1800.00	428.76	0.071
Total	3600.00	857.52	2700.00	643.14	2400.00	571.68	3000.00	714.60	0.867	3600.00	857.52	2400.00	571.68	1800.00	428.76	0.071
Developmental Cost																
Domestic Helper	4200.00	1000.44	5100.00	1214.82	3600.00	857.52	6000.00	1429.20	0.655	3600.00	857.52	6000.00	1429.20	-	-	0.020 *
Nursery	0.00	0.00	0.00	0.00	0.00	0.00	2400.00	571.68	0.400	0.00	0.00	1200.00	285.84	1200.00	285.84	0.326
Special Education	0.00	0.00	0.00	0.00	0.00	0.00	600.00	142.92	0.555	0.00	0.00	0.00	0.00	1200.00	285.84	0.485
Dietary Supplements	5700.00	1357.74	5400.00	1286.28	6000.00	1429.20	6000.00	1429.20	0.987	6000.00	1429.20	6000.00	1429.20	3600.00	857.52	0.520
Daily Necessities	2400.00	571.68	2400.00	571.68	3300.00	786.06	4500.00	1071.90	0.300	2400.00	571.68	2700.00	643.14	3300.00	786.06	0.881
Insurance	120.00	28.58	1250.00	297.75	1790.00	426.38	1300.00	309.66	0.549	1250.00	297.75	1740.00	414.47	-	-	0.503
Others	0.00	0.00	1200.00	285.84	1200.00	285.84	0.00	0.00	0.582	1200.00	285.84	1200.00	285.84	0.00	0.00	0.308
Total	8520.00	2029.46	12,000.00	2858.40	14,400.00	3430.08	15,600.00	3715.92	0.235	13,200.00	3144.24	15,600.00	3715.92	9300.00	2215.26	0.334
Indirect Cost																
Parents and Child Productivity Loss	15,000.00	3573.00	32,400.00	7717.68	24,000.00	5716.80	54,000.00	12862.80	0.032 *	14,400.00	3430.08	35,400.00	8432.28	27,600.00	6574.32	0.090
Total	15,000.00	3573.00	32,400.00	7717.68	24,000.00	5716.80	54,000.00	12862.80	0.032 *	27,900.00	6645.78	34,100.00	8122.62	26,700.00	6359.94	0.090
Overall Total	30,720.00	7317.50	52,360.00	12,472.15	45,400.00	10,814.28	78,000.00	18579.60	0.299	49,500.00	11,790.90	57,520.00	13,701.26	39,600.00	9432.72	0.189

^a^*p*-Value for the Kruskal–Wallistest. * Significant *p*-Value. CP = Cerebral Palsy. RM = Ringgit Malaysia. USD = United States Dollar.

**Table 4 ijerph-19-00475-t004:** Comparison of CP economic burden according to children’s sex, household income, and severity based on GMFCS level.

Economic Burden	Sex					Household Income					Severity (Based on GMFCS Level)						
	Male		Female		Mann-Whitney U ^b^	Below Average		Above Average		Mann-Whitney U ^b^	GMFCS Level 3		GMFCS Level 4		GMFCS Level 5		Kruskal-Wallis ^a^
	RM	USD	RM	USD		RM	USD	RM	USD		RM	USD	RM	USD	RM	USD	
Direct Healthcare Cost																	
Rehabilitation Services	1200.00	285.84	2100.00	500.22	0.954	1920.00	457.34	0.00	0.00	0.234	2400.00	571.68	600.00	142.92	1200.00	285.84	0.369
Alternative Treatments	600.00	142.92	1200.00	285.84	0.789	1200.00	285.84	0.00	0.00	0.730	1200.00	285.84	0.00	0.00	600.00	142.92	0.562
Medicine Costs	0.00	0.00	300.00	71.46	0.912	0.00	0.00	0.00	0.00	0.983	600.00	142.92	1275.00	303.71	0.00	0.00	0.861
Purchase of Medical Aids	3125.00	744.38	1560.00	371.59	0.257	2365.00	563.34	6800.00	1619.76	0.031 *	3500.00	833.70	0.00	0.00	2750.00	655.05	0.396
Diagnostic Tests	0.00	0.00	0.00	0.00	0.169	0.00	0.00	0.00	0.00	0.785	0.00	0.00	0.00	0.00	0.00	0.00	0.838
Ward Entrance Fees/Surgery	0.00	0.00	0.00	0.00	0.795	0.00	0.00	0.00	0.00	0.481	0.00	0.00	0.00	0.00	0.00	0.00	0.402
Consultation Fees	0.00	0.00	0.00	0.00	0.770	0.00	0.00	0.00	0.00	0.497	0.00	0.00	-	-	0.00	0.00	0.406
Total	4925.00	1173.14	5160.00	1229.11	0.941	5485.00	1306.53	6800.00	1619.76	0.817	7700.00	1834.14	1875.00	446.63	4550.00	1083.81	0.132
Direct Non-Healthcare Cost																	
Transportation Cost	3000.00	714.60	2400.00	571.68	0.788	3300.00	786.06	2400.00	571.68	0.219	3000.00	714.60	3600.00	857.52	2400.00	571.68	0.133
Total	3000.00	714.60	2400.00	571.68	0.788	3300.00	786.06	2400.00	571.68	0.219	3000.00	714.60	3600.00	857.52	2400.00	571.68	0.133
Developmental Cost																	
Domestic Helper	5100.00	1214.82	3600.00	857.52	0.272	4500.00	1071.90	7800.00	1857.96	0.392	4800.00	1143.36	3600.00	857.52	6000.00	1429.20	0.331
Nursery	0.00	0.00	0.00	0.00	0.910	540.00	128.63	0.00	0.00	0.098	3000.00	714.60	0.00	0.00	600.00	142.92	0.186
Special Education	0.00	0.00	0.00	0.00	0.780	0.00	0.00	0.00	0.00	0.792	1200.00	285.84	0.00	0.00	0.00	0.00	0.665
Dietary Supplements	6000.00	1429.20	4800.00	1143.36	0.980	6000.00	1429.20	5700.00	1357.74	0.927	4800.00	1143.36	4800.00	1143.36	6000.00	1429.20	0.241
Daily Necessities	3600.00	857.52	2400.00	571.68	0.298	3000.00	714.60	2400.00	571.68	0.757	2400.00	571.68	2400.00	571.68	3600.00	857.52	0.302
Insurance	1000.00	238.20	1740.00	414.47	0.815	1000.00	238.20	2140.00	509.75	0.321	1340.00	319.19	1500.00	357.30	1400.00	333.48	0.863
Others	1200.00	285.84	1200.00	285.84	0.612	1200.00	285.84	0.00	0.00	0.457	0.00	0.00	1200.00	285.84	1200.00	285.84	0.197
Total	16,900.00	4025.58	13,740.00	3272.87	0.143	16,240.00	3868.37	18,040.00	4297.13	0.160	17,540.00	4178.03	13,500.00	3215.70	18,800.00	4478.16	0.513
Indirect Cost																	
Parents and Child Productivity Loss	28,800.00	6860.16	27,000.00	6431.40	0.922	24,000.00	5716.80	93,000.00	22,152.60	0.018 *	31,800.00	7574.76	30,000.00	7146.00	24,000.00	5716.80	0.918
Total	28,800.00	6860.16	27,000.00	6431.40	0.922	24,000.00	5716.80	93,000.00	22,152.60	0.018 *	31,800.00	7574.76	30,000.00	7146.00	24,000.00	5716.80	0.918
Overall Total	51,600.00	12,291.12	45,500.00	10,838.10	0.699	46,350.00	11,040.57	120,164.00	28,623.06	0.264	57,050.00	13,589.31	50,860.00	12,114.85	45,400.00	10,814.28	0.424

^a^*p*-Value for the Kruskal–Wallistest. ^b^*p*-Value for the Mann–Whitney U test. * *p* < 0.05 taken as level of significance. RM = Ringgit Malaysia. USD = United States Dollar.

**Table 5 ijerph-19-00475-t005:** Factors associated with the total economic burden of parents/caregivers of children with CP in the study population (n = 106).

Variable	SLR ^a^				MLR ^b^				
	B ^c^	(95% CI) ^e^		*p*-Value	Adjusted b ^d^	(95% CI) ^e^		*t*-Statistic	*p*-Value
(Constant) ^f^								3.45	0.001
Annual Household Income (RM)	0.25	0.06	0.43	0.010 *	0.23	0.04	0.42	2.39	0.019 *
Parents’ Age Group ^g^	11,845.87	−172.35	23,864.09	0.053	0.14	−3418.51	20,956.06	1.43	0.157
Child’s Sex ^h^	5515.95	−7366.12	18,398.01	0.398	0.13	−4026.09	21,175.55	1.35	0.180
Severity—GMFCS Level ^i^	−8200.53	−27,320.59	10,919.53	0.397	−0.11	−29,359.79	7936.47	−1.14	0.257

^a^ Simple linear regression(SLR). ^b^ Multiple linear regression (MLR) (R^2^ = 0.105; The model fits reasonably well; model assumptions are met; no interaction between independent variables; no multicollinearity problem). ^c^ Crude regression coefficient. ^d^ Adjusted regression coefficient. ^e^ 95% CI = 95% confidence interval (lower and upper bounds). ^f^ (Constant) = for multiple linear regression, with *B* = 24,163.96. ^g^ Parents’ age group = 36–55 years old. ^h^ Child’s sex = Male. ^i^ Severity—GMFCS Level = Level 3. * *p* < 0.05 taken as significance level.

## Data Availability

The data presented in this study are available on request from the corresponding author. The data are not publicly available due to some restrictions, and they are only available on reasonable request.

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
