# Peer review of "The Economic Burden and Determinant Factors of Parents/Caregivers of Children with Cerebral Palsy in Malaysia: A Mixed Methods Study"

_ijerph, 2022, doi:10.3390/ijerph19010475_

Round 1
Reviewer 1 Report
Dear authors, thank you for your submission. I think it is an interesting manuscript and is generally well-written and presented. I think more detail is needed in some parts and some work is needed on the discussion.
Some specific comments are given below. Please also note that the manuscript should be edited as there are some grammatical and editing errors present.
2.3 some more explanation around the tool development and validation process should be included.
Examples of the semi-structured questions can also be given.
2.7 a reference and some description on the qualitative process used can be included here. For example, which theorist or approach for thematic analysis was used? Were transcripts of the interviews transcribed verbatim? How were the themes identified?
3.4 Data in the form of quotes is usually provided for qualitative findings to show credibility of your findings. Quotes can be included here under each main finding, or given in a table, or given as supplementary material if the manuscript is too long.
4. Discussion
A lot of the results are repeated here, and I think the discussion needs rewriting. I suggest you avoid repeating results (eg percentages and RM) and focus on the issues, eg with child age and example statement that reflects the issues could be -“most children were boys, which reflects the global prevalence of gender and CP”. I also suggest you start with your most important results that answers your research question. For example, the aim was to explore the economic burden on parents/caregivers of a child with CP, so the important result is best given first – did you answer your research question? What unique contribution/s do your study results make to the field?
As it is a mixed method study there should be some linkage of the qualitative data with the quantitative data – does the qualitative data support the quantitative data? This should be clear in the discussion.
Reviewer 2 Report
Thank you for the opportunity to review the current article about the factors that determine the economic burden in parents of children with CP. The topic is interesting, especially given that it analyzes the economic cost in Malaysia, were not many research have been done.
However, the articles present serious flaws and much work is needed to modify the whole structure of the paper.
Major Points
- In the introduction (and in other parts of the article) the inclusion of RM income is difficult to follow and interpret, so it will be necessary to include the equivalence with dollars so readers can understand the impact of the results.
- More information about the specific Malaysia context is needed. Which are the socioeconomic characteristics of the country that may have an impact on families that have a children diagnosed with a disability?
- At the end of the introduction authors outline a number of factors that may influence economic burden. Please explain them and review the previous research that has identified this factors in other countries or contexts.
- It is not clear to me why authors decide to use a mixed-method design. Despite the objectives of the quantitative part are stated, there is no information about the research questions for the qualitative part.
- Why were some participants from the quantitative part selected to participate in the qualitative interview? Did authors use any kind of criteria to select them?
- There is an important lack of information regarding the study instruments. What is the name of the “validated questionnaire”? Is there any data regarding validity, reliability or factorial structure that can support that the questionnaire is validated? What type of questions did you use in the semi-structured interview?
- In the inclusion criteria, what does it mean that “severe CP”? We don’t know if parents were aware of the real state of their children. In addition, there is no section of limitations at the end of the discussion section.
- In the study variables sections there is no indications of the qualitative aspects of the research, only the quantitative variables. Again, we can argue why authors decide to do a qualitative research after the quantitative.
- More justification is needed about why authors decide to categorize age in groups of three, two or and six years (0-3; 4-6; 7-12 and >13). This fact should be based on previous research or on some theoretical background. The same for the parents’ age.
- There is no information about the procedure to conduct the qualitative interviews. I’ll recommend to use the COREQ guidelines for reporting qualitative results. In addition, the qualitative analysis is not fully explained, authors just indicate that they performed a thematic analysis, but with no references to the paradigms or to the type of TA employed.
- The results section is too long and it’s easy to get lost with so many data. It will be better to present the main statistically significant results and the post-hoc comparisons.
- The qualitative results section doesn’t include any quotations or verbatim from participants, so it is impossible to compare if the codes and themes identified really are in line with the parents’ discourse.
- In the qualitative results section I think you’re trying to explain the main themes, not the “items”. Please clarify.
- The discussion section should begin with the aim of the research and the main results. Authors introduce and describe again some of the results, but there is a lack of comparisons with previous research, and with comparisons with what happen in other countries.
Minor points:
- Include some data regarding the qualitative part in the abstract
- In tables 3 and 4 the data from the Kruskal Wallis test is the p-value? Please indicate this information.
Reviewer 3 Report
The article
The economic burden and determinants of parents/carers of children with Cerebral Palsy in Malaysia: Mixed Methods Study, addresses a topic of great importance to the welfare of the disabled child, in this case Cerebral Palsy. It is very well structured and very well written and reasoned. It presents an adequate and very well informed methodology and produces conclusions of reference for the improvement of the welfare system in Malaysia and the deepening of public policies of protection and conceived in a transversal, interdisciplinary and collaborative logic.
I only leave as a suggestion to the authors:
- Integrate in the introduction a paragraph on the social protection system, education system and health system in Malaysia in order to allow a better understanding of the results, their discussion and the conclusions.
In the conclusions I also suggest the writing of a paragraph based on the results proposing the need for the promotion of more integrative public policies from the perspective of the person.
Round 2
Reviewer 1 Report
Dear authors, thank you. I have reviewed the revised version and believe it has addressed the feedback comments I previously provided. You have done a lot of work on the manuscript and I believe it is a more detailed and a lot clearer. Check the final manuscript for grammatical and editing errors to ensure it complies with academic writing conventions.
Reviewer 2 Report
Authors have really improved the quality of the manuscript and the qualitative findings are now more in line with a mixed-method design. I think that with all this modifications the article can be published.